# Floristic Composition and Species Conservation Status in Three *Polylepis* (Rosaceae) Relict Forests in Peru

**DOI:** 10.3390/plants14223537

**Published:** 2025-11-20

**Authors:** Yakov Quinteros-Gómez, Jehoshua Macedo-Bedoya, Flavia Anlas-Rosado, Sergio Yangua-Evangelista, Franco Angeles-Alvarez, Shirley Azurín-Sotelo, Marcel La Rosa-Sánchez, Doris Gómez-Ticerán, Enoc Jara-Peña, José Campos-De la Cruz, Bruno Padilla-Torres, Iván Fernández-De la Cruz

**Affiliations:** 1Laboratorio de Ecología Tropical y Análisis de Datos, Facultad de Ciencias Biológicas, Universidad Nacional Mayor de San Marcos (UNMSM), Lima 15001, Peru; yquinterosg@unmsm.edu.pe (Y.Q.-G.); jehoshua.macedo@unmsm.edu.pe (J.M.-B.); flavia.anlas@unmsm.edu.pe (F.A.-R.); sergio.yangua@unmsm.edu.pe (S.Y.-E.); franco.angeles@unmsm.edu.pe (F.A.-A.); shirley.azurin@unmsm.edu.pe (S.A.-S.); sandro.larosasanchez@unmsm.edu.pe (M.L.R.-S.); exequiel.padilla@unmsm.edu.pe (B.P.-T.); ivan.fernandez1@unmsm.edu.pe (I.F.-D.l.C.); 2Grupo de Investigación MOCA, Facultad de Ciencias Matemáticas, Universidad Nacional Mayor de San Marcos (UNMSM), Lima 15001, Peru; dgomezt@unmsm.edu.pe; 3Laboratorio de Fitología Aplicada, Facultad de Ciencias Biológicas, Universidad Nacional Mayor de San Marcos (UNMSM), Lima 15001, Peru; 4Herbario San Marcos, Universidad Nacional Mayor de San Marcos (UNMSM), Lima 15001, Peru; jocamde@gmail.com; 5Departamento de Ecología, Museo de Historia Natural, Universidad Nacional Mayor de San Marcos (UNMSM), Lima 15001, Peru

**Keywords:** anthropization, conservation, diversity, fragmentation, queñua

## Abstract

This study assessed the floristic composition, endemism, and conservation status of three relict *Polylepis* forests (“queñua”) in Oyón, Peru, located between 4000 and 4500 m a.s.l. A total of 150 vascular plant species were recorded, belonging to 100 genera and 47 families. Asteraceae (52 spp.) and Poaceae (17 spp.) were the most diverse, while herbs dominated the life-form spectrum (73%). Species richness decreased significantly with elevation (R^2^ = 0.86, *p* < 0.001): Zone 3 (4000 m) hosted 100 species, Zone 2 had 52, and Zone 1 had 35. Floristic similarity (Jaccard index) revealed that Zone 3 was the most distinct, sharing as little as 6% of its species with higher zones. Endemism was notable, with 14 species (9.33%), 100% concentrated in Zone 3, which also harbored 71.4% of the 7 species categorized as threatened, including *Polylepis incana* (CR) and Senecio nutans (VU). These findings identify lower-elevation relicts as critical biodiversity hotspots, likely sustained by milder microclimates and greater habitat heterogeneity. However, ongoing threats from fragmentation, logging, and grazing jeopardize their persistence. Therefore, site-specific conservation strategies that prioritize these lower-altitude hotspots are urgently required.

## 1. Introduction

The Andes constitute the longest active mountain range on Earth [1], stretching approximately 7000 km across the continent of South America [2]. This mountain range provides livelihoods and essential ecosystem services to millions of people [3]. However, its origin is still under debate; the most widely accepted theories hold that its formation dates back to the Late Cretaceous, around 80 million years ago [4] as a result of the subduction of the Nazca and Caribbean oceanic plates beneath the South American continental plate [1,2]. The Andes host a remarkable diversity of native and endemic species [5], which are distributed across a series of ecosystems such as paramos, grasslands, wetlands, glaciers, and shrublands, among others [6]. Their distribution is determined by changes in climatic conditions (temperature, solar radiation, wind exposure) and by the altitudinal gradient characteristic of the Andean region [7].

The genus *Polylepis* Ruiz & Pav. (Rosaceae), endemic to the mountains of South America [8], comprises approximately 27 species of trees and shrubs [8,9,10,11], commonly known as *queñua* or *queñoa*. These species dominate the canopy of forests and shrublands whose conservation and restoration are considered priorities at a continental level [12]. The genus is distributed along the tropical and subtropical Andes of South America, from Venezuela to northern Chile and Argentina [8,13,14] between 3500 and 5200 m a.s.l. [8,10]. This broad distribution has also fueled debate regarding the irregular distribution of the *Polylepis* forest (often disconnected from the treelines of lower-altitude forests by puna grasslands) [15], a pattern mainly attributed to climatic variations and the intensification of human activities [16], which represent a major threat to biodiversity, particularly in global hotspots like the Tropical Andes [17].

Abiotic factors such as bioclime, precipitation, solar exposure, substrate type and characteristics, as well as hydrological conditions, often limit the growth of seedling, forest development and recovery [18,19]. In addition, the upward shift of the altitudinal limit of the human activities has been identified as a key factor in the reduction in their extent, as well as in their degradation (wood internal structure) and in the limited success of conservation efforts [20,21].

*Polylepis* forests play an important role in the dynamics of ecosystem processes at high altitudes [22]. They reduce soil erosion, retain sediments and nutrients [23], and provide organic matter to soils, enriching them and increasing their volume and water absorption capacity [24]. Furthermore, the characteristic morphology of individuals of this genus, such as its dense branching and small leaves [8] enhances water input as their leaves and branches act as a network that captures moisture from fog, a common phenomenon in many mountains where these forests occur [24]. This capacity for water input is also related to the high diversity of mosses associated with these forests, which form a continuous layer on the soil and branches, intercepting rainfall and contributing to moisture retention during the dry season [25]. These features facilitate the formation of diverse microhabitats that harbor a rich floristic diversity, including endemic and threatened species [26]. Moreover, *Polylepis* forests act as carbon sinks [27,28], protect water sources [29], regulate runoff, improve the quality of water flowing into rivers and streams [24] and provide essential resources for local communities [10] by serving as valuable sources of medicine, food, and fuel [15,24].

*Polylepis* species are characterized by their small, imparipinnate leaves and reddish trunks with thin, exfoliating bark [30]. The reproductive phase occurs predominantly during the dry and cold season, while vegetative growth develops during the humid season with higher temperature [10]. Peru, the country with the greatest diversity of the genus, is represented by approximately 23 species [11,31], distributed across Andean territories marked by pronounced environmental heterogeneity, notable species endemism, and the presence of distinctive phenotypic adaptations [32], resulting from the extreme ecological conditions characteristic of the high-Andean environment.

In the provinces of Cajatambo, Yauyos, and Oyón (Lima, Peru), *Polylepis* forests are found [26,33,34]. These forests face topographic challenges due to the rugged orography, which has significantly restricted research in these areas and generated substantial information gaps across extensive territories. The aim of the present study was to develop a comprehensive inventory of vascular plants associated with three queñua (*Polylepis)* forests under different levels of human activity in the province of Oyón, Peru.

## 2. Materials and Methods

### 2.1. Study Area

The study was conducted between September 2023 and August 2024 in three relict forests of queñua (*Polylepis* spp.) located in the Huaura River basin, Oyón Province, Lima Department, Peru; at elevations between 4000 and 4500 m a.s.l. (Figure 1). These forests experience variable climatic conditions with mean temperatures ranging from 4.2 °C to 15.8 °C, and receive an average of 547 mm of annual precipitation [35].

The first evaluated forest, Zone 1 (Z1), is located at coordinates 10°35′22.8″ S, 76°50′21.9″ W (WGS84), between 4350 and 4400 m a.s.l., and is characterized by a landscape dominated by herbaceous vegetation with scattered shrubs (Figure 2A). The second forest, Zone 2 (Z2), is located at 10°34′21.2″ S, 76°50′36.1″ W, at 4500 m a.s.l., in an area near the Raura Mine, and was highly fragmented due to the construction of the Oyón-Cajatambo road (Figure 2B). The third forest, Zone 3 (Z3), located at 10°36′03.1″ S, 76°48′52.7″ W, at 4000 m a.s.l., is crossed by the Ushpa River and the same road, which determines the existence of two distinct sectors (upper and lower) with evidence of recent anthropogenic disturbance through logging and firewood extraction (Figure 2C).

### 2.2. Collection Method

For the floristic inventory, intensive surveys were conducted over three consecutive days in each forest, covering both the dry and wet seasons to maximize the recording of species in their reproductive stage. Due to the rugged topography and the fragmented nature of the forests, an intensive search method using directed walks (“random-walk”) was chosen to more effectively cover habitat heterogeneity. The survey was carried out by a team of four researchers moving in parallel, separated by 5 m, to ensure homogeneous coverage of the study areas. A total sampling effort of 144 person-hours was invested in each of the three forests, corresponding to three days of work with 6 h of effective searching per person each day, in both dry and wet seasons.

Specimens found were photographed in situ, and those that could not be identified in the field were collected in duplicate. The botanical material was processed following standard herbarium techniques, identified using taxonomic keys, specialized databases, and by comparison with reference specimens, and finally deposited in the collection of the Laboratory of Tropical Ecology and Data Analysis at the Universidad Nacional Mayor de San Marcos, under collection numbers YQG 342–387. Nomenclature followed the criteria of World Flora Online [36].

### 2.3. Conservation Status

The conservation status of the registered species was assessed using two classification systems: the Red List of the International Union for Conservation of Nature (IUCN) [37] and the Peruvian classification established in Supreme Decree No. 043–2006–AG (Classification of Threatened Wild Flora Species) [38]. This dual assessment made it possible to determine the level of threat to each taxon according to national and international criteria.

### 2.4. Statistical Analyses

Statistical analyses were performed in RStudio version 4.3.3 [39], mainly using the vegan package version 2.7-2 for diversity and similarity analyses, the stats package version 4.0.6 for correlation tests, dendextend version 1.19.1 for dendrogram construction, and ggplot2 version 4.0.1 [40] for visualization of results.

To assess floristic similarity among study zones, the Jaccard dissimilarity index was employed using the vegdist(method = “jaccard”) function, which ranges from 0 (identical composition) to 1 (completely dissimilar composition). The results of the Jaccard dissimilarity matrix were visualized through a heat map generated with the pheatmap package version 1.0.13 [41]. Additionally, a dendrogram was constructed using hierarchical clustering based on the dissimilarity matrix, utilizing the hclust(method = “average”) function from the stats package version 4.0.6 [39] and as.dendrogram(…) from the dendextend package [42], to graphically represent similarity relationships among zones.

An analysis of species distribution by family was performed using a heat map that visualized the specific richness in each study area. To do this, a custom function was developed in R that grouped the data by family and area, filtering out those families with at least two recorded species. The resulting matrix was visualized with the pheatmap package version 1.0.13 [41]. This analysis allowed us to identify the dominant families in each area and compare the taxonomic distribution patterns among the three *Polylepis* forests.

Correlation analyses were conducted to explore general patterns between key environmental and floristic variables. The relationship between altitude and the total number of recorded species was examined using Pearson’s correlation function (cor.test(x, y, method = “pearson”)), considering the mean elevation of each study zone (Z1: 4375 m, Z2: 4500 m, Z3: 4000 m) and the corresponding species counts. This analysis was applied with descriptive purposes, as an exploratory approach to visualize the direction and strength of the observed pattern along the altitudinal gradient. All statistical procedures were performed in R, adopting a significance level of α = 0.05 to maintain consistency with the analytical framework used for the other variables.

## 3. Results

A total of 150 vascular plant species were recorded, distributed across 100 genera and 47 families (Table 1). The distribution by life form showed a predominance of herbs with 110 species (73.33%), followed by shrubs with 29 species (19.33%), vines with 4 species (2.67%), succulents and trees with 3 species (2%), and one stoloniferous species (0.67%) (Figure 3A). The most diverse families were Asteraceae (52 species), Poaceae (17 species), Fabaceae (5 species), Calceolariaceae (4 species), Solanaceae (4 species) and Polypodiaceae (4 species), which together accounted for 57.3% of the total species (Figure 3B). The best represented genera were *Baccharis* L. (8 species), *Senecio* L. (7 species), *Ageratina* Spach (5 species), and *Calceolaria* L. and *Cinnagrostis* Griseb. (4 species each) (Figure 3C).

A gradual decrease in the number of recorded species was observed with increasing elevation (Figure 4A). Zone 3 (4000 m a.s.l.) exhibited the highest floristic diversity with 100 species, followed by Zone 2 (4500 m a.s.l.) with 52 species and Zone 1 (4375 m a.s.l.) with 35 species. The fitted linear trend illustrates a consistent pattern of declining diversity toward higher altitudes, highlighting the ecological influence of elevation on the distribution of plant assemblages across the *Polylepis* forest gradient.

This pattern is further supported by diversity indices (Figure 4B), which demonstrate a similar altitudinal trend. The Shannon index (H’) increased from 3.56 in Z1 to 4.61 in Z3, while the Simpson index rose from 0.97 to 0.99, indicating not only a greater number of species but also a more even distribution of abundances in the lower-elevation forest. These values suggest that Z3 sustains a more heterogeneous and balanced community, with a relatively lower dominance of particular taxa compared to the upper zones.

In forest Z1 (4375 m a.s.l.), 35 species distributed across 29 genera and 8 families were recorded. The distribution by life forms showed 27 herbs (77.14%), 6 shrubs (17.14%), 1 tree (2.86%), and 1 succulent (2.86%). The most diverse families were Asteraceae (27 species) and Poaceae (2 species), jointly representing 82.85% of the total species in the zone. The most diverse genera were *Senecio* (4 species) and *Baccharis* (3 species). Figure 5 illustrates some of the species recorded in the study area.

In forest Z2 (4500 m a.s.l.), 52 species grouped into 37 genera and 17 families were recorded. The distribution by life forms included 42 herbs (80.77%), 6 shrubs (11.54%), 2 succulents (3.85%), 1 tree (1.92%), and 1 vine (1.92%). The most diverse families were Asteraceae (25 species) and Poaceae (8 species), representing 63.46% of the total species recorded in the zone. The most diverse genera were *Baccharis* (5 species), *Ageratina* (4 species), *Cinnagrostis* (4 species), and *Senecio* (4 species).

In forest Z3 (4000 m a.s.l.), the highest floristic diversity was recorded with 100 species grouped into 73 genera and 42 families. The distribution by life forms showed 69 herbs (69%), 23 shrubs (23%), 4 vines (4%), 2 trees (2%), 1 succulent (1%), and 1 stoloniferous herb (1%). The most diverse families were Asteraceae (25 species), Poaceae (9 species), Fabaceae (5 species), and Solanaceae (4 species), representing 43% of the total species recorded in the zone. The most diverse genera were *Calceolaria* (4 species), followed by *Ageratina*, *Baccharis*, *Gynoxys*, *Ophryosporus* Meyen, *Plantago* L., *Senecio*, and *Solanum* L. (3 species each). 

The heat map analysis (Figure 6) reveals contrasting patterns of taxonomic diversity among zones. Asteraceae exhibits the highest intensity of species richness across all three zones, being particularly dominant in Z1, where it reaches maximum diversity values. In Z2 and Z3, Asteraceae maintains high richness but with intermediate values.

Poaceae shows a differential distribution pattern, with higher intensity in Z2 and Z3 compared to Z1, where it presents very low values. The families Fabaceae, Calceolariaceae, and Solanaceae appear exclusively in Z3 with moderate intensities, suggesting greater functional diversity in the lowest elevation zone.

The lateral dendrogram of the heat map groups families according to their distribution patterns, showing two main clusters: one composed of widely distributed families (Asteraceae and Poaceae) and another of families with more restricted distribution. Most families with low species richness are concentrated in higher elevation zones (Z1 and Z2), while Z3 presents the greatest diversity both in number of families and in species richness per family.

The floristic overlap among the three study zones was first visualized using a Venn diagram (Figure 7). This analysis revealed a high number of unique species in the lowest elevation zone (Z3) and a low overall similarity among the sites. Specifically, Zone 3 contained 82 exclusive species, while the higher-elevation forests, Zone 1 and Zone 2, had only 14 and 23 exclusive species, respectively. The greatest overlap was observed between these two higher-altitude sites, which shared 13 species.

To quantify the degree of this floristic dissimilarity, a Jaccard index was calculated (Figure 8A). The dissimilarity matrix confirmed that Zone 3 was the most distinct zone, with high dissimilarity values relative to Zone 1 (0.94) and Zone 2 (0.88). In contrast, Zone 1 and Zone 2 showed a lower dissimilarity value of 0.72, indicating a greater floristic similarity between them.

A hierarchical clustering dendrogram further validated this pattern (Figure 8B). The dendrogram shows a clear separation between Zone 3, which forms an independent branch, and the other two zones (Z1 and Z2), which cluster together in a common clade. This topology reflects that the two higher-elevation forests share more floristic characteristics with each other than they do with the distinct, lower-elevation forest.

Endemic taxa (14) are strongly concentrated in Zone 3 (4000 m a.s.l.), which hosts the largest share of narrow-range species recorded in the survey (Table 1). However, endemism is not strictly exclusive to that site: for example, *Gnaphalium dombeyanum* occurs across all three zones and *Gynoxys nitida* is present in both Z1 and Z3. Figure 9A shows a clear positive association between total recorded species and number of endemic taxa, indicating that sites with higher overall richness tend to harbor more endemics. The observed distribution (Figure 9B) (a majority of endemics in the lower-elevation patch together with a few taxa distributed across multiple zones) suggests that Zone 3 functions as a primary reservoir of local endemism while some endemic species exhibit broader ecological tolerance and distribution within the gradient.

A total of 22 species were identified with a recognized conservation status under national (Supreme Decree No. 043-2006-AG) and international (IUCN) classifications (Figure 10). According to the IUCN Red List, 19 species were assessed, most of which are categorized as Least Concern (LC), including *Baccharis latifolia*, *B. nitida*, *Culcitium canescens*, *Senecio condimentarius*, *Taraxacum officinale*, *Berberis lutea*, *Austrocylindropuntia floccosa*, *Cystopteris fragilis*, *Ephedra americana*, *Pernettya prostrata*, *Eucalyptus globulus*, *Poa annua*, *Cantua buxifolia*, *Monnina salicifolia*, *Polylepis incana*, *Solanum lanceolatum*, and *S. nitidum*. Two species were classified as Near Threatened (NT): *Aristeguietia discolor* and *Polylepis weberbaueri*.

Under the Peruvian legal framework (Supreme Decree No. 043-2006-AG), five species were listed in threat categories: *Polylepis incana* (Critically Endangered), *Polylepis weberbaueri* (Vulnerable), *Senecio nutans* (Vulnerable), *Cantua buxifolia* (Near Threatened), and *Ephedra americana* (Near Threatened). These taxa represent approximately 4.7% of the total recorded flora, showing a distinct concentration of threatened species in Zone 3 (4000 m a.s.l.), where five of the seven listed taxa occur. *Polylepis incana*, *S. nutans*, *C. buxifolia*, and *E. americana* are all restricted to this lower-elevation site, while *P. weberbaueri* is present in both Zones 1 and 2.

Several taxa listed under national (Supreme Decree No. 043-2006-AG) and IUCN categories concentrate in Zone 3, reinforcing its conservation importance for both endemic and threatened flora. Notable examples include *Polylepis incana* (CR under DS), *Senecio nutans* (VU), *Cantua buxifolia* (NT) and *Ephedra americana* (NT), while *Polylepis weberbaueri* occurs in the higher patches (Z1 and Z2).

## 4. Discussion

*Polylepis* forests represent a unique biological system in the Andes, with great ecological and biogeographical interest [11,43,44]. However, they are recognized as one of the high Andean ecosystems with the highest threat level in South America [45], which endangers their ecological integrity [46,47]. Our results, which identified 150 vascular plant species in three small relict forests, reveal that these ecosystems in the province of Oyón harbor significant floristic diversity despite facing these pressures. The predominance of herbaceous life forms (73.3%) over shrubs and trees (Figure 3A) is consistent with patterns described for other high Andean ecosystems, where herbs exhibit greater adaptive plasticity to extreme environmental conditions such as frost, high solar radiation, and thin soils [48].

The floristic composition documented in these small *Polylepis* forest patches (most under 5 ha) revealed a remarkably high diversity compared with other *Polylepis* forests across the Peruvian Andes. Studies conducted in the Lima region have reported forest relicts dominated by several *Polylepis* species [11,26,33,49], which aligns with our findings of *P. incana* and *P. weberbaueri* as the prevailing canopy species. While an outstanding 282 species have been recorded in the larger forests of Yauyos (>30 ha) [26], floristic surveys in the Vilcanota mountain range, the southern Andes of Ayacucho, and Otishi National Park have reported between 76 and 178 species [44,50,51,52]. In this context, the 150 species documented in the present study illustrate that even small and fragmented *Polylepis* remnants can sustain a floristic richness comparable to larger forest systems. This observation reinforces the ecological importance of these relicts as key reservoirs of regional biodiversity, where structural complexity and microhabitat variation contribute to maintaining diverse plant assemblages, even under conditions of isolation [53].

The altitudinal gradient clearly acts as a primary driver of floristic patterns, generating a mosaic of environmental conditions that shape species composition [54,55,56]. Our finding of a significant decline in species richness with elevation (Figure 4) is consistent with broad-scale patterns observed along both wet paramo gradients in Ecuador and drier puna gradients in Bolivia and northern Chile, where harsh climatic conditions at higher altitudes act as a strong environmental filter [57,58,59]. This environmental filtering is also evident in the floristic dissimilarity among our sites (Figure 7 and Figure 8). The high distinctiveness of Zone 3 (4000 m a.s.l.) suggests its position at a critical ecotonal boundary, likely incorporating elements from lower-elevation montane shrublands that are filtered out from the true high-puna environments of Zone 1 and Zone 2. The clustering of the two higher-altitude forests reflects their shared exposure to the more extreme conditions characteristic of the puna, thereby supporting a more specialized and less diverse subset of the regional flora. This pattern of species turnover underscores the importance of altitudinal gradients in shaping the transition between Andean vegetation belts. This process of compositional shift is being accelerated by climate change-induced thermophilization, whereby cold-adapted, narrow-niched species are progressively replaced by more tolerant, generalist taxa, leading to a loss of structural and compositional differentiation [55,59,60,61].

The taxonomic composition was dominated by the Asteraceae family (52 species) (Figure 3B), a finding consistent with its rank as the second most species-rich family in Peru and its well-documented success in high-Andean ecosystems [62,63]. The success of this family in these ecosystems is attributed to adaptive traits such as pubescent or succulent leaves and deep roots, which allow them to thrive in extreme conditions [34]. The high representation of genera like *Baccharis* (8 spp.), *Senecio* (7 spp.), and *Ageratina* (5 spp.) (Figure 3C) also aligns with their known role as effective colonizers in disturbed high-Andean environments [64,65]. Consistent with other high Andean floristic inventories, Asteraceae and Poaceae were the dominant families [49,66,67,68,69,70]. Our study confirmed the dominance of Asteraceae across the entire gradient, while Poaceae (17 species) was more prominent at lower elevations, an altitudinal alternation that has been documented in previous research [71].

From a conservation perspective, our most critical finding is the concentration of endemic and threatened species in the lowest-elevation forest, Zone 3. This site functions as a biodiversity hotspot, harboring 100% of the recorded endemic species and 71.4% of threatened taxa, including the Critically Endangered *Polylepis incana* (Figure 9 and Figure 10; Table 1). The strong positive association between total species richness and endemism (Figure 9) suggests that the underlying drivers promoting high overall diversity also sustain rare and specialized taxa. This concentration can likely be attributed to a combination of three factors. First, the milder and more stable microclimate at 4000 m a.s.l., with a lower frequency of extreme frosts, provides more favorable conditions for the persistence of narrowly distributed species that are less tolerant to climatic extremes [72]. Second, the higher habitat heterogeneity in Zone 3, which is intersected by the Ushpa River, creates unique riparian microhabitats with greater soil moisture and nutrient availability, conditions not present in the drier, slope-based habitats of Z1 and Z2; such riparian zones are well-known to act as local biodiversity refuges [73]. Third, the disturbance regime may differ in intensity and history. While the impacts of logging and grazing in Zone 3 appear recent, the fragmentation affecting Zone 2 is associated with long-standing mining activity, likely resulting in a more profound and historically integrated disturbance to the ecosystem. These insights have direct implications for local management and restoration: conservation efforts must be site-specific, prioritizing the legal protection of these lower-elevation hotspots. Restoration priorities should focus on securing riparian corridors within these zones, controlling selective logging, and implementing grazing exclusion zones, rather than applying a uniform strategy across the entire gradient. These findings reinforce the ecological and conservation significance of lower-elevation *Polylepis* relicts, demonstrating that even small patches can harbor disproportionately high biological value and should therefore be prioritized.

This high conservation value, however, faces significant threats. It is estimated that the original extent of queñua forests has been reduced to less than 10% in the highlands of Bolivia and Peru [29]. This reduction is mainly due to anthropogenic activities driven by economic, social, and cultural factors [30,44], such as burning [51,74], logging [51,75], and replacement by exotic species [76]. The limited research on these impacts constitutes a significant limitation for informed decision-making [31,77,78], compromising the implementation of effective climate change adaptation strategies [30,79].

Our research complements previous regional inventories by providing a novel analysis of how floristic composition and conservation status shift across an altitudinal gradient within Oyón’s *Polylepis* ecosystems. While this comparative approach is the study’s key contribution, it is important to acknowledge that with three study sites, it is challenging to completely isolate the effect of altitude from other site-specific variables, like soil type and specific land-use history. Expanding this gradient analysis to a larger number of sites is a next step to further strengthen the relationship between elevation and floristic patterns.

## 5. Conclusions

The relict Polylepis forests of the Oyón province represent critical high-Andean ecosystems that sustain exceptional floristic diversity, with 150 species recorded across the study sites. The observed variation along the altitudinal gradient reflects a clear ecological structuring, where the lowest-elevation forest (Z3, 4000 m a.s.l.) stands out as a key biodiversity reservoir. This site concentrated 100% of the recorded endemic taxa and more than 70% of the threatened species, including the Critically Endangered Polylepis incana. These findings highlight that lower-elevation relicts play a disproportionately important role in preserving regional diversity and endemism within fragmented Andean landscapes.

The distinctiveness of this lower-altitude hotspot, likely sustained by milder microclimates and greater habitat heterogeneity associated with its riparian setting, makes it a non-negotiable priority for conservation. This study, therefore, calls for site-specific management strategies that move beyond general protection. Rather than a uniform approach, restoration efforts should focus on securing these lower-altitude riparian corridors and addressing local threats such as selective logging and grazing. Strengthening local stewardship and promoting targeted research on ecological connectivity and population viability will be crucial to ensure the long-term resilience of these irreplaceable *Polylepis* forests and the ecosystem services they provide.

## Figures and Tables

**Figure 1 plants-14-03537-f001:**
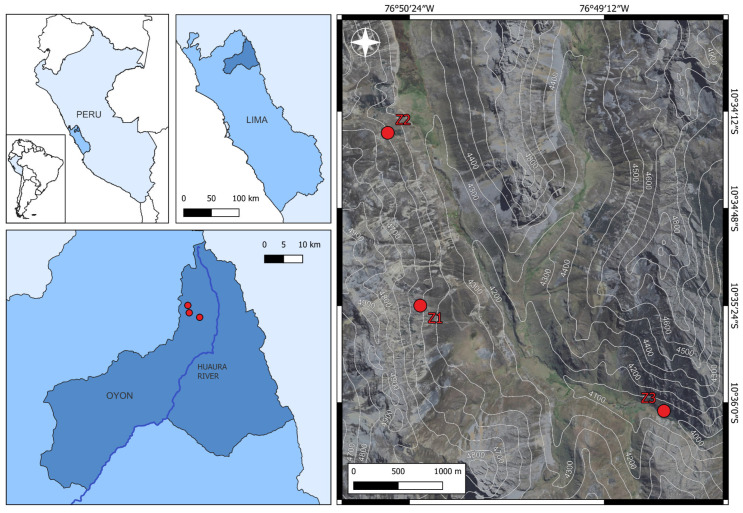
Location of the three *Polylepis* forests studied in the Oyón province, Peru.

**Figure 2 plants-14-03537-f002:**
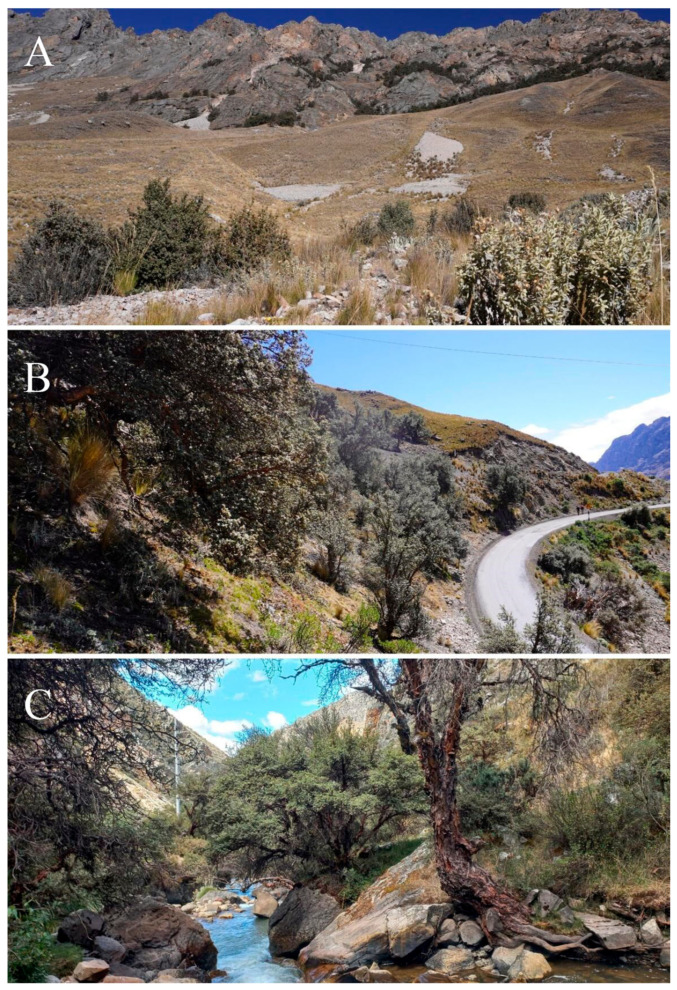
The three study zones in the province of Oyón. (**A**) Zone 1 (4375 m a.s.l.). (**B**) Zone 2 (4500 m a.s.l.). (**C**) Zone 3 (4000 m a.s.l.). Photographs by Jehoshua Macedo-Bedoya.

**Figure 3 plants-14-03537-f003:**
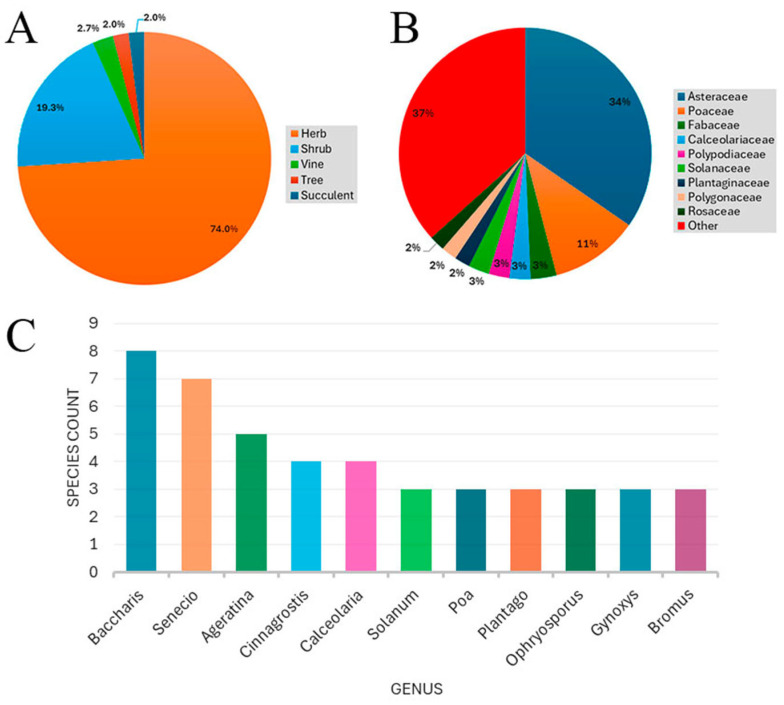
(**A**) Number of species according to life forms. (**B**) Number of species by botanical families. (**C**) Number of species of the most abundant genera.

**Figure 4 plants-14-03537-f004:**
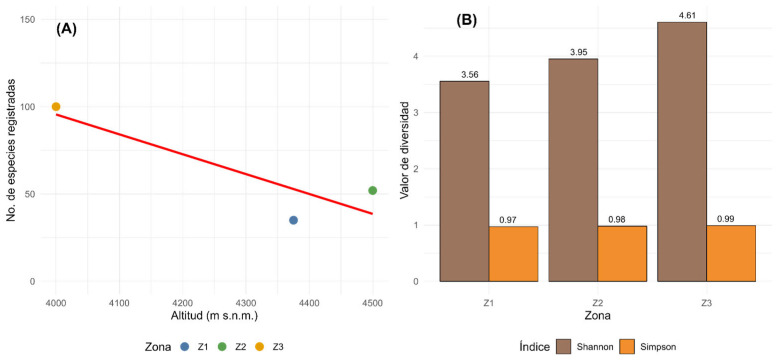
(**A**) Altitudinal gradient effect on the number of recorded species across the three study zones. (**B**) Diversity indices (Shannon and Simpson) of the three study zones.

**Figure 5 plants-14-03537-f005:**
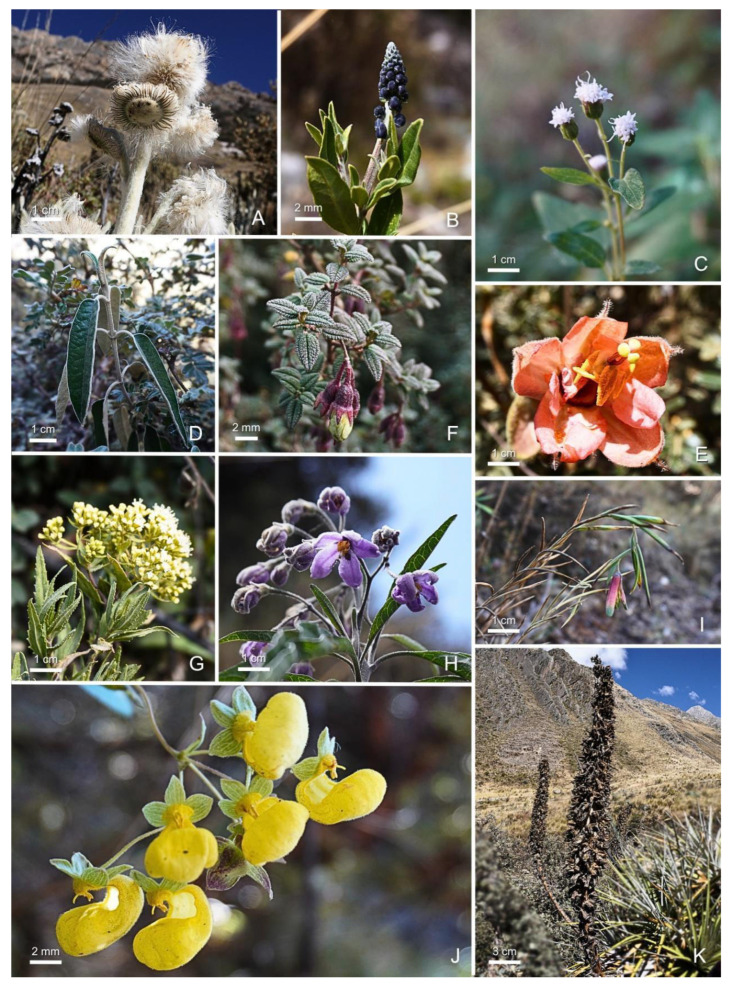
Selected species associated with *Polylepis* in the Oyón province. (**A**) *Culcitium canescens*, (**B**) *Monnina salicifolia*, (**C**) *Aristeguietia discolor*, (**D**) *Gynoxys oleifolia*, (**E**) *Passiflora mixta*, (**F**) *Brachyotum ledifolium*, (**G**) *Ophryosporus chilca*, (**H**) *Solanum nitidum*, (**I**) *Bomarea dulcis*, (**J**) *Calceolaria* sp., (**K**) *Puya alpestris*. Photographs by Yakov Quinteros-Gómez and Jehoshua Macedo-Bedoya.

**Figure 6 plants-14-03537-f006:**
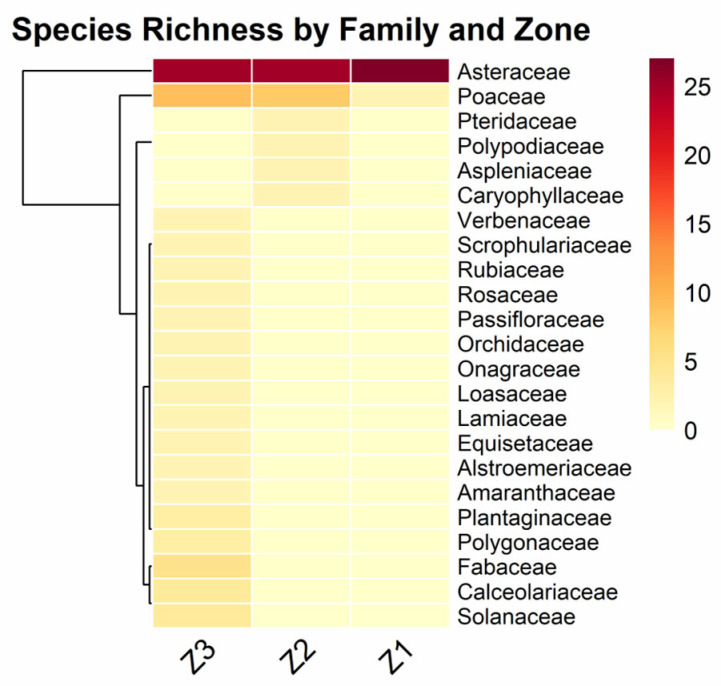
Number of recorded species distribution by botanical families across the three *Polylepis* forest zones.

**Figure 7 plants-14-03537-f007:**
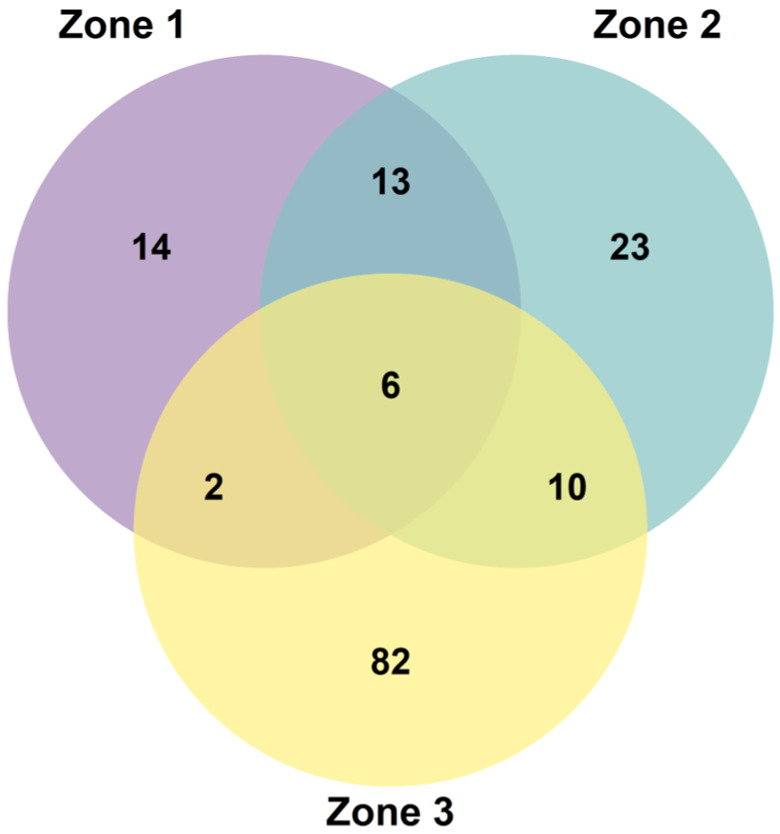
Venn diagram showing the number of shared and exclusive plant species among the three studied *Polylepis* forests.

**Figure 8 plants-14-03537-f008:**
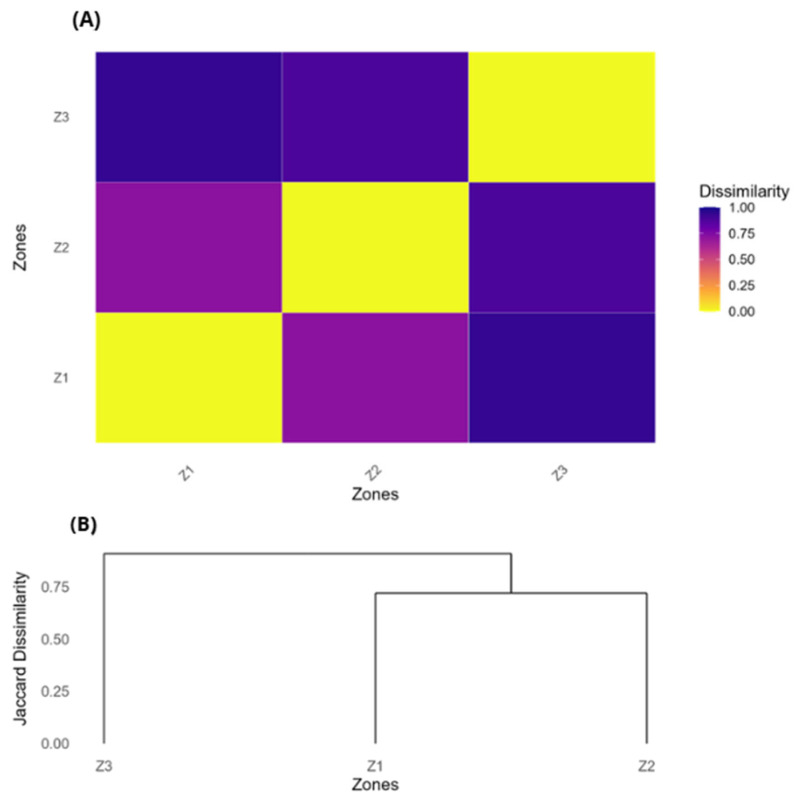
Quantitative analysis of floristic dissimilarity. (**A**) Jaccard dissimilarity matrix heatmap displaying pairwise dissimilarity values between zones. (**B**) Hierarchical clustering dendrogram showing relationships between study zones.

**Figure 9 plants-14-03537-f009:**
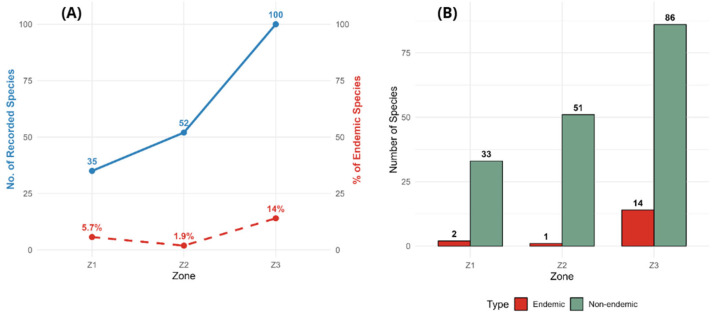
(**A**) Relationship between total species diversity and endemic species richness across study zones. (**B**) Comparison of the absolute number of endemic and non-endemic species in each zone.

**Figure 10 plants-14-03537-f010:**
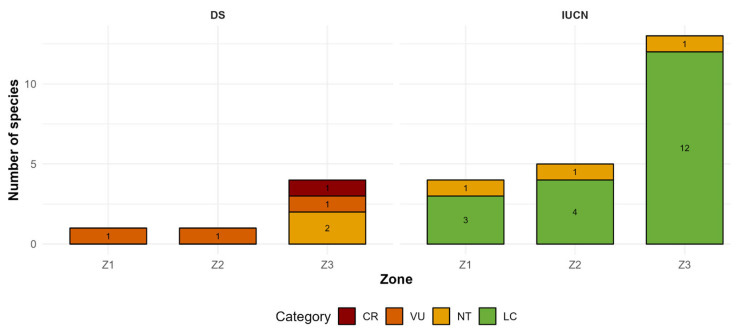
Distribution of threatened species by study zones according to Peruvian Supreme Decree categories (CR: Critically Endangered, VU: Vulnerable, NT: Near Threatened) and IUCN Red List categories (NT: Near Threatened, LC: Least Concern).

**Table 1 plants-14-03537-t001:** List of reported species in the evaluated *Polylepis* forests.

Family	Species	Growth Habit	*IUCN*/SD *	Endemic	Zones
Alstroemeriaceae	*Bomarea dulcis* (Hook.) Beauverd	Vi	-	No	Z2, Z3
	*Bomarea ovata* (Cav.) Mirb.	Vi	-	No	Z3
Amaranthaceae	*Alternanthera lanceolata* (Benth.) Schinz	H	-	No	Z3
	*Alternanthera macbridei* Standl.	H	-	No	Z3
Aspleniaceae	*Asplenium peruvianum* Desv.	H	-	No	Z2
	*Asplenium triphyllum* C. Presl	H	-	No	Z2, Z3
Asteraceae	*Achyrocline alata* (Kunth) DC.	H	-	No	Z1, Z2, Z3
	*Achyrophorus taraxacoides* Walp.	H	-	No	Z1, Z2, Z3
	*Ageratina azangaroensis* (Sch. Bip. ex Wedd.) R.M. King & H. Rob.	H	-	No	Z2, Z3
	*Ageratina glechonophylla* (Less.) R.M. King & H. Rob.	H	-	No	Z1, Z3
	*Ageratina pentlandiana* (DC.) R.M. King & H. Rob.	H	-	No	Z2
	*Ageratina sternbergiana* (DC.) R.M. King & H. Rob.	H	-	No	Z2, Z3
	*Ageratina* sp.	H	-	-	Z2
	*Aristeguietia discolor* (DC.) R.M. King & H. Rob.	H	*NT*	No	Z3
	*Baccharis alpina* Kunth	B	-	No	Z1, Z2
	*Baccharis caespitosa* (Ruiz & Pav.) Pers.	B	-	No	Z1, Z2
	*Baccharis latifolia* (Ruiz & Pav.) Pers.	B	*LC*	No	Z3
	*Baccharis nitida* (Ruiz & Pav.) Pers.	B	*LC*	No	Z3
	*Baccharis salicina* Torr. & A. Gray	B	-	No	Z3
	*Baccharis* sp.	B	-	-	Z2
	*Baccharis tola* Phil.	B	-	No	Z1, Z2
	*Baccharis tricuneata* (L. f.) Pers.	B	-	No	Z2
	*Belloa* sp.	H	-	-	Z1, Z2
	*Bidens andicola* Kunth	H	-	No	Z2, Z3
	*Calendula officinalis* L.	H	-	No	Z3
	*Chaptalia nutans* (L.) Polák	H	-	No	Z1
	*Chersodoma antennaria* (Wedd.) Cabrera	H	-	No	Z1, Z2
	*Conyza bonariensis* (L.) Cronquist	H	-	No	Z3
	*Conyza canadensis* (L.) Cronquist	H	-	No	Z1
	*Culcitium canescens* Bonpl.	H	*LC*	No	Z1
	*Diplostephium* sp.	H	-	-	Z1
	*Gnaphalium dombeyanum* DC.	H	-	Yes	Z1, Z2, Z3
	*Gynoxys nitida* Muschl.	B	-	Yes	Z1, Z3
	*Gynoxys oleifolia* Muschl.	B	-	Yes	Z3
	*Gynoxys visoensis* Cuatrec.	B	-	Yes	Z3
	*Hieracium* sp.	H	-	-	Z1
	*Hypochaeris chillensis* (Kunth) Britton	H	-	No	Z1
	*Laennecia artemisiifolia* (Meyen & Walp.) G.L. Nesom	H	-	No	Z2
Asteraceae	*Lasiocephalus* sp.	H	-	-	Z1
	*Loricaria ferruginea* (Ruiz & Pav.) Wedd.	B	-	No	Z1
	*Onoseris odorata* Hook. & Arn.	H	-	Yes	Z3
	*Ophryosporus chilca* (Kunth) Hieron.	B	-	No	Z3
	*Ophryosporus heptanthus* (Sch. Bip. ex Wedd.) R.M. King & H. Rob.	B	-	No	Z3
	*Ophryosporus piquerioides* (DC.) Benth. ex Baker	B	-	No	Z1, Z2, Z3
	*Paranephelius ovatus* A. Gray ex Wedd.	H	-	No	Z1, Z2
	*Rockhausenia nubigena* Kunth	H	-	No	Z1, Z2
	*Senecio comosus* Sch. Bip.	H	-	No	Z1, Z2, Z3
	*Senecio condimentarius* Cabrera	H	*LC*	No	Z1, Z2
	*Senecio crassiflorus* DC.	H	-	No	Z2
	*Senecio evacoides Sch. Bip.*	H	-	No	Z2, Z3
	*Senecio hohenackeri Sch. Bip.*	H	-	No	Z1
	*Senecio nutans* Sch. Bip.	H	**VU**	No	Z3
	*Senecio sublutescens* Cuatrec.	H	-	No	Z1
	*Sonchus asper* (L.) Hill	H	-	No	Z3
	*Sonchus oleraceus* L.	H	-	No	Z3
	*Taraxacum officinale* F.H. Wigg.	H	*LC*	No	Z2
	*Werneria villosa* A. Gray	H	-	No	Z1, Z2
	*Xenophyllum* sp.	H	-	No	Z1
Berberidaceae	*Berberis lutea* Ruiz & Pav.	B	*LC*	No	Z3
Brassicaceae	*Weberbauera spathulifolia* (A. Gray) O.E. Schulz	H	-	No	Z1, Z2
Bromeliaceae	*Puya alpestris* (Poepp.) Gay	StH	-	No	Z3
Cactaceae	*Austrocylindropuntia floccosa* (Salm-Dyck ex Winterfeld) F.Ritter	Suc	*LC*	No	Z1, Z2
Calceolariaceae	*Calceolaria glauca* Ruiz & Pav.	H	-	Yes	Z3
	*Calceolaria hispida* Benth.	H	-	Yes	Z3
	*Calceolaria parvifolia* Wedd.	H	-	No	Z3
	*Calceolaria* sp.	H	-	-	Z3
Campanulaceae	*Lobelia decurrens* Cav.	H	-	No	Z3
Caryophyllaceae	*Stellaria weddellii* Pedersen	H	-	No	Z2
	*Paronychia andina* A. Gray	H	-	No	Z2
Cyperaceae	*Cyperus* sp.	H	-	-	Z3
Cystopteridaceae	*Cystopteris fragilis* (L.) Bernh.	H	*LC*	No	Z2
Dryopteridaceae	*Polystichum cochleatum* (Klotzsch) Hieron.	H	-	No	Z2
	*Polystichum orbiculatum* (Desv.) J. Rémy & Fée	H	-	No	Z3
Ephedraceae	*Ephedra americana* Humb. & Bonpl. ex Willd.	H	*LC*/**NT**	No	Z3
Equisetaceae	*Equisetum bogotense* Kunth	H	-	No	Z3
	*Equisetum* sp.	H	-	-	Z3
Ericaceae	*Pernettya prostrata* (Cav.) DC.	B	*LC*	No	Z3
Fabaceae	*Astragalus* sp.	H	-	-	Z2, Z3
	*Lupinus brachypremnon* C.P. Sm.	B	-	Yes	Z3
	*Lupinus condensiflorus* C.P. Sm.	B	-	Yes	Z3
	*Otholobium pubescens* (Poir.) J.W. Grimes	B	-	No	Z3
	*Senna birostris* (Dombey ex Vogel) H.S. Irwin & Barneby	B	-	Yes	Z3
Geraniaceae	*Geranium* sp.	H	-	-	Z3
Juncaceae	*Luzula racemosa* Desv.	H	-	No	Z1, Z2, Z3
Lamiaceae	*Clinopodium sericeum* (C. Presl ex Benth.) Govaerts	B	-	Yes	Z3
	*Lepechinia meyenii* (Walp.) Epling	H	-	No	Z3
Loasaceae	*Caiophora cirsiifolia* C. Presl	H	-	Yes	Z3
	*Loasa* sp.	H	-	-	Z3
Loranthaceae	*Tristerix pubescens* Kuijt	B	-	Yes	Z3
Malvaceae	*Malva* sp.	H	-	-	Z3
Melastomataceae	*Brachyotum ledifolium* (Desr.) Triana	B	-	No	Z3
Myrtaceae	*Eucalyptus globulus* Labill.	T	*LC*	No	Z3
Onagraceae	*Oenothera rosea* L’Hér. ex Aiton	H	-	No	Z3
	*Oenothera laciniata* Hill	H	-	No	Z3
Orchidaceae	*Aa paleacea* (Kunth) Rchb. f.	H	-	No	Z3
	*Altensteinia fimbriata* Kunth	H	-	No	Z3
Orobanchaceae	*Castilleja* sp.	H	-	-	Z1
	*Neobartsia melampyroides* (Kunth) Uribe-Convers & Tank	H	-	No	Z3
Oxalidaceae	*Oxalis megalorrhiza* Jacq.	H	-	No	Z3
Passifloraceae	*Passiflora mixta* L. f.	Vi	-	No	Z3
	*Passiflora trifoliata* Cav.	Vi	-	No	Z3
Piperaceae	*Peperomia microphylla* Kunth	Suc	-	No	Z3
	*Peperomia galioides* Kunth	Suc	-	No	Z2
Plantaginaceae	*Plantago australis* Lam.	H	-	No	Z3
	*Plantago lamprophylla* Pilg.	H	-	No	Z3
	*Plantago* sp.	H	-	-	Z3
Poaceae	*Aciachne pulvinata* Benth.	H	-	No	Z3
	*Agrostis tolucensis* Kunth	H	-	No	Z2, Z3
	*Bothriochloa* sp.	H	-	-	Z3
	*Bromus catharticus* Vahl	H	-	No	Z3
	*Bromus pitensis* Kunth	H	-	No	Z2
	*Bromus* sp.	H	-	-	Z3
	*Cinnagrostis heterophylla* (Wedd.) P.M. Peterson, Soreng, Romasch. & Barberá	H	-	No	Z2
	*Cinnagrostis intermedia* (J. Presl) P.M. Peterson, Soreng, Romasch. & Barberá	H	-	No	Z2
	*Cinnagrostis tarmensis* (Pilg.) P.M. Peterson, Soreng, Romasch. & Barberá	H	-	No	Z1, Z2
	*Cinnagrostis vicunarum* (Wedd.) P.M. Peterson, Soreng, Romasch. & Barberá	H	-	No	Z2
	*Festuca humilior* Nees & Meyen	H	-	No	Z3
	*Festuca myuros* L.	H	-	No	Z2
	*Jarava ichu* Ruiz & Pav.	H	-	No	Z3
	*Muhlenbergia peruviana* Meisn.	H	-	No	Z1
	*Poa annua* L.	H	*LC*	No	Z3
	*Poa gilgiana* Pilg.	H	-	No	Z3
	*Poa macusaniensis* (E.H.L. Krause) Refulio	H	-	No	Z2
Polemoniaceae	*Cantua buxifolia* Juss. ex Lam.	B	*LC/* **NT**	No	Z3
Polygalaceae	*Monnina salicifolia* Ruiz & Pav.	H	*LC*	Yes	Z3
Polygonaceae	*Muehlenbeckia volcanica* (Benth.) Endl.	H	-	No	Z2, Z3
	*Rumex crispus* L.	H	-	No	Z3
Polygonaceae	*Rumex obtusifolius* L.	H	-	No	Z3
Polypodiaceae	*Campyloneurum angustifolium* (Sw.) Fée	H	-	No	Z3
	*Pleopeltis pycnocarpa* (C. Chr.) A.R. Sm.	H	-	No	Z1
	*Polypodium* sp.	H	-	-	Z2
	*Thelypteris* sp.	H	-	-	Z2
Pteridaceae	*Cheilanthes pruinata* Kaulf.	H	-	No	Z2, Z3
	*Cheilanthes pilosa* Goldm.	H	-	No	Z2
Ranunculaceae	*Ranunculus praemorsus* Kunth ex DC.	H	-	No	Z3
Rosaceae	*Polylepis incana* Kunth	T	*LC/* **CR**	No	Z3
	*Polylepis weberbaueri* Pilg.	T	*NT/* **VU**	No	Z1, Z2
	*Rubus* sp.	B	-	-	Z3
Rubiaceae	*Arcytophyllum* sp.	H	-	-	Z3
	*Galium hypocarpium* (L.) Endl. ex Griseb.	H	-	No	Z3
Scrophulariaceae	Scrophulariaceae sp.	H	-	-	Z3
	*Buddleja* sp.	B	-	-	Z3
Solanaceae	*Dunalia spinosa* (Meyen) Dammer	B	-	No	Z3
	*Solanum lanceolatum* Cav.	H	*LC*	No	Z3
	*Solanum nitidum* Ruiz & Pav.	H	*LC*	No	Z3
	*Solanum* sp.	H	-	-	Z3
Urticaceae	*Urtica magellanica* Juss. ex Poir.	H	-	No	Z3
Verbenaceae	*Verbena litoralis* Kunth	H	-	No	Z3
	*Duranta* sp.	B	-	-	Z3
Woodsiaceae	*Woodsia* sp.	H	-	-	Z2

Reported species categorized according to their growth habit: (**H**: herbaceous; **B**: Shrub; **Suc**: succulent; **Vi**: vine; **T**: tree; **StH**: stoloniferous herbaceous) and their distribution across three altitudinal zones (Z1, Z2, and Z3). * *IUCN* = International Union for Conservation of Nature, **SD** = Supreme Decree No. 043-2006-AG.

## Data Availability

Data used in this study can be requested to the following authors via email: yquinterosg@unmsm.edu.pe, jehoshua.macedo@unmsm.edu.pe, flavia.anlas@unmsm.edu.pe.

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
