# Peer review of "Floristic Composition and Species Conservation Status in Three Polylepis (Rosaceae) Relict Forests in Peru"

_plants, 2025, doi:10.3390/plants14223537_

Round 1

Reviewer 1 Report

Comments and Suggestions for Authors

dear authors,

I appreciate the work done in a wonderful global place. In my opinion the manuscript presented is well written and argued, the figures are excellent as the table presented. However the work needs to improve with some integrations, crucial for international audience, as Plant journal.

The authors propose a manuscript titled “Floristic composition and species conservation status in three Polylepis (Rosaceae) relict forests in Peru”. Particular attention was given on topic aspect on new wild plant species updating the floristic composition, endemism, and conservation status of three relict Polylepis forests in the belt mountain of Peru. The flora is dominated of many families, especially Asteraceae and Poaceae, while the Endemism have a high number of taxa as Polylepis incana and Senecio nutans that are both listed in the IUCN. Crucial notices are given on Conservation strategies and ecological restoration. I appreciate the original idea of the work which with a few revisions will convince me and the editor to publish it on Journal.

  1. Introduction

Please add a references in the few points that I have indicated because these are already known concepts, and in some cases complete (bold font) in the suggested way. Check whole introduction in this way.

  • Line 51. “The genus Polylepis (Rosaceae), endemic to the mountains of South America [choose reference]”;
  • Lines 59-60. a pattern mainly attributed to climatic variations and the intensification of human activities, that a global level especially in the hot spot plant biodiversity [choose reference];
  • Lines 61-63. Abiotic factors such as bioclime, precipitation, solar exposure, substrate type and chemical soil characteristics [Ben Mahmoud et al.2024], as well as hydrological conditions [17,18], often limit the growth of seedling, forest development and recovery [choose reference];
  • Lines 70-72. Furthermore, the characteristic morphology of individuals of this genus [choose reference] of enhances water input [choose reference], as their leaves and branches act as a network that captures moisture from fog, a common phenomenon in many mountains where these forests occur [23];
  • Lines 80. Being by serving as valuable sources of medicine, food, and fuel [15,23, Perrino et al. 2024].

Reference to be added:

  • Ben Mahmoud et al. 2024. Euro-Mediterranean Journal for Environmental Integration. https://doi.org/10.1007/s41207-024-00468-5
  • Perrino et al. 2024. Planta. https://doi.org/10.1007/s00425-024-04571-3

  1. Materials and Methods
  • The authors declare that the field sampling were done in only two mounths (maybe less?), but probably there are other plant species in other seasons! right? If yes please complete the information that the species blooming in other season are not in the list;
  • Figure 1 and lines 106-107, 109, 111-112. Please complete in the text the geographical system used in the georeferenced map. WGS84? Gauss-Boaga? ….. Please specify
  • Figure 2. Very wonderful pictures, but please add the author of the photo;
  • The authors decalre that have studied 69 points with floristic diversity…. but with what criteria were chosen the 69 plot sampling? to cover all the vegetation environments in the area? Please explain better and how many times did you return to the same plot?
  • Line 131-134. Please about official herbarium complete the information with herbarium name and code. See index herbariorum: https://sweetgum.nybg.org/science/ih/

  1. Results

Wel done, some suggetstions.

  • When reported the genus or species as scientific name for the first time in the text, please report the name in the complete way, see https://powo.science.kew.org as:
  • Baccharis

Also for others, and check whole manuscript

  • Senecio ….
  • Ageratina ….
  • Calceolaria ….
  • Cinnagrostis ….

  1. Conclusion

Please a few more words about the future prospects of research

Regard,

reviewer

Author Response

Introduction

Comments:

  • Line 51. “The genus Polylepis (Rosaceae), endemic to the mountains of South America [choose reference]”;
  • Lines 59-60. a pattern mainly attributed to climatic variations and the intensification of human activities, that a global level especially in the hot spot plant biodiversity [choose reference];
  • Lines 61-63. Abiotic factors such as bioclime, precipitation, solar exposure, substrate type and chemical soil characteristics [Ben Mahmoud et al.2024], as well as hydrological conditions, often limit the growth of seedling, forest development and recovery [choose reference];
  • Lines 70-72. Furthermore, the characteristic morphology of individuals of this genus [choose reference] of enhances water input [choose reference], as their leaves and branches act as a network that captures moisture from fog, a common phenomenon in many mountains where these forests occur;
  • Lines 80. Being by serving as valuable sources of medicine, food, and fuel [15,23, Perrino et al. 2024].

Response:
We thank the reviewer for these constructive suggestions to strengthen the Introduction. We have, accordingly, revised the text as follows:

  • Line 51: We have added an appropriate reference for this statement.
  • Lines 59-60: We have expanded the sentence to connect our statement to the broader concept of biodiversity hotspots and have added a relevant reference.
  • Lines 61-63: We think that the ideas were already supported by their respective references, but we have reorganize and clarified this sentence.
  • Lines 70-72: We have clarified this sentence and ensured that both the morphology and the function of water input are supported by appropriate references.
  • Line 80: We think that the ideas were already supported by their respective references.

Material and Methods

Comments: Figure 1. and lines 106-107, 109, 111-112. Please complete in the text the geographical system used in the georeferenced map. WGS84? Gauss-Boaga? ….. Please specify
Response: Agree. We have now specified the geographic coordinate system (WGS84) in the manuscript, as requested. This can be found in the Study Area section.

Comments: Figure 2. Very wonderful pictures, but please add the author of the photo;
Response: We agree. A photo credit has now been added to the legend of Figure 2.

Comments: The authors decalre that have studied 69 points with floristic diversity…. but with what criteria were chosen the 69 plot sampling? to cover all the vegetation environments in the area? Please explain better and how many times did you return to the same plot?
Response: We believe there may be a slight misunderstanding, as our study was based on intensive surveys within three distinct forest zones, rather than a design using 69 plots. This comment may have been intended for a different paper. However, we have re-read our Collection method section to ensure its clarity and confirm that our sampling design is described in sufficient detail.

Comments: Line 131-134. Please about official herbarium complete the information with herbarium name and code. See index herbariorum: https://sweetgum.nybg.org/science/ih/
Response: We thank the reviewer for this important point on specimen deposition. We wish to clarify that the botanical material was deposited in the institutional collection of the Laboratory of Tropical Ecology and Data Analysis at UNMSM, which does not have an official Index Herbariorum code. To enhance rigor and traceability, we have now specified the exact collection numbers (YQG & JMB 342–387) in the Collection method section.

Results

Comments: Wel done, some suggetstions. When reported the genus or species as scientific name for the first time in the text, please report the name in the complete way...
Response: We agree. The manuscript has been systematically revised to include the author authority for each taxon at its first mention in the main text.

Conclusion

Comments: Please a few more words about the future prospects of research
Response: We thank the reviewer for this suggestion. We agree that this strengthens the conclusion, and we have accordingly expanded the final paragraph of the manuscript to include a more detailed outlook on future research prospects. 

Reviewer 2 Report

Comments and Suggestions for Authors

plants-3936691

Floristic composition and species conservation status in three Polylepis (Rosaceae) relict forests in Peru

While I strongly believe that floristic treatments have  to be more widely broadcast in scientific journals, I found that the paper dealt with only three rather similar forest plots, and has too many tables with similar information. For example, the diagrams about IUCN threatened species are not meaningful and could easily be supressed in favour of a more complete species list that has plant habit, threat level and presence/absence in one of the 3 study plots. The IUCN categories of theat are VU, EN and CR, with little to be gained listing LC or NT as well. These could be jointly shown in a circular diagram rather than in bar diagrams, however just stating how many species are in each category in the list and in total would be sufficient.

Table 1 and Table 2 can be merged and presented as the results of the ms.

To study only three plots is not enough to provide meaningful phytosociology analyses – especially if the work does not involve abundance or density, only species presence and absence. The similarities and differences between three plots can be glimpsed in a venn diagram the overlap and differences between species composition, as well as number of singletons.

Author Response

We sincerely thank the reviewer for their detailed and insightful comments, which have helped us to significantly improve the clarity and presentation of our manuscript.

We agree with the reviewer’s suggestion to streamline the presentation of our data and reduce redundancy. Accordingly, we have merged the previous Tables 1 and 2 into a single, comprehensive species list (now Table 1). As recommended, this new table now includes detailed information on plant habit, conservation status, and presence/absence in each of the three study zones. We have also undertaken a thorough review of all figures and have replaced or reformatted several with higher-resolution versions to enhance their clarity and readability.

Regarding the suggestions on alternative analytical figures, we found the reviewer's idea of using a Venn diagram to be particularly insightful. We have, therefore, added a Venn diagram (now Figure 7) to the manuscript. As suggested, this figure provides a clear and intuitive visualization of the floristic overlap and the number of unique species (singletons) for each zone.

Concerning the figures for threatened species (now Figure 9), we have retained the bar charts as we feel they effectively and immediately emphasize the disproportionate concentration of threatened species in Zone 3, which is a key conclusion of our manuscript. However, as per the reviewer’s suggestion, all the detailed information is now also available in the new comprehensive Table 1.

Reviewer 3 Report

Comments and Suggestions for Authors

While the text is clear, it would benefit from language polishing to improve fluency and reduce repetition (especially in Results and Discussion).

The correlation between altitude and species richness (R² = 0.86, p < 0.001) is based on only three sites (n = 3). Although the authors acknowledge this, reviewers may expect a stronger justification or a more cautious interpretation.

Some figures (especially heatmaps) might be difficult to interpret without color legends or clearer captions. Ensure high-resolution and well-labeled visuals for submission.

Minor typographical issues and inconsistencies (spacing, capitalization of species names, etc.) should be corrected before submission.

Author Response

We thank the reviewer for their thorough and constructive feedback, which has helped us to improve the quality and clarity of our manuscript.

We agree with the reviewer's assessment regarding the need for language refinement. Accordingly, the manuscript has undergone a comprehensive language polishing and proofreading process, with a focus on the Results and Discussion sections to improve fluency and correct minor typographical issues.

Regarding the interpretation of the correlation analysis (n=3), we appreciate the reviewer's call for a more cautious interpretation. While our Methods section already acknowledged the small sample size, we have now revised the Discussion to further emphasize the descriptive nature of this finding. We have rephrased our interpretation to frame the strong correlation not as definitive proof, but as a compelling pattern consistent with well-established ecological theory.

We also agree that the clarity of our figures is essential. We have therefore undertaken a full review of all visual elements. All figures, including the heatmaps, have been re-exported at a higher resolution, and we have added clear color legends and revised all figure captions to be more descriptive, as suggested.

Reviewer 4 Report

Comments and Suggestions for Authors

The manuscript "Floristic composition and species conservation status in three Polylepis (Rosaceae) relict forests in Peru" presents a valuable floristic inventory and conservation assessment of three Polylepis forests in the province of Oyón, Peru. The work is timely and relevant given the critical conservation status of Polylepis ecosystems in the Andes. The manuscript is clearly written and follows the journal’s structure, with comprehensive data on species composition, life forms, and conservation categories. However, some methodological and interpretive aspects need refinement before the paper can be accepted.

Major Comments

Sampling Design and Representativeness: The study design relies on three sites along an altitudinal gradient, but the sampling effort per site and its comparability should be clarified. Although the “random-walk” approach is practical in rugged terrain, it lacks statistical rigor for comparing richness across sites. I suggest to include area-based sampling effort or plot-equivalent area (e.g., species per 100 m²) to support quantitative comparisons.

Statistical Analyses: The correlation between altitude and species richness (R² = 0.86, p < 0.001) is robust, but with only three data points (n = 3), the test lacks inferential validity. Please rephrase this as a descriptive trend rather than a statistically significant correlation. Additionally, the authors could consider rarefaction or diversity indices (Shannon, Simpson) to better capture within-site heterogeneity.

Floristic Composition and Elevational Gradient: The discussion correctly highlights the decline in richness with elevation, but it would benefit from integration with known altitudinal patterns of Andean vegetation (e.g., treeline dynamics, páramo-puna transitions). Consider citing comparative floristic gradients from Ecuador, Bolivia, or northern Chile.

Conservation Implications: The paper convincingly demonstrates that Zone 3 (lowest elevation) concentrates endemic and threatened species. However, the discussion should explicitly address why—is this due to milder climate, soil fertility, or lower disturbance intensity? Incorporating local management perspectives or restoration priorities would strengthen the applied value.

Figures and Tables: Figures 6–9 are informative but dense. Simplify labels and ensure legends are self-explanatory. Table 1 could be condensed into a supplementary file to improve readability.

Minor Comments

Materials and Methods: Clarify whether all zones belong to the same watershed.

Results: When referring to families (Asteraceae, Poaceae), indicate percentage contribution for clarity.

Discussion: The conclusion that small relicts act as biodiversity refuges is important—emphasize potential integration into regional conservation plans.

References: Verify DOIs and consistency in journal names (e.g., Rev. peru. biol.Revista Peruana de Biología).

Overall Recommendation

Minor to Moderate Revision

The study is scientifically sound and contributes meaningful new data on Polylepis forest floristics and conservation. Addressing methodological transparency, refining the interpretation of statistical results, and tightening the language will substantially improve the manuscript’s impact and clarity.

Comments on the Quality of English Language

The manuscript is generally well written but requires editing for conciseness and consistent English grammar. For example: Replace “showed a marked fragmentation” with “was highly fragmented.”

Avoid redundancy (“the most diverse families were Asteraceae (52 species) and Poaceae (17 species)… together accounting for 57.3 % of the total species” can be shortened).

Author Response

Reviewer Comment 1 - Sampling Design and Representativeness

"The study design relies on three sites along an altitudinal gradient, but the sampling effort per site and its comparability should be clarified. Although the “random-walk” approach is practical in rugged terrain, it lacks statistical rigor for comparing richness across sites. I suggest to include area-based sampling effort or plot-equivalent area (e.g., species per 100 m²) to support quantitative comparisons."

Authors response 1

We thank the reviewer for this point on methodological rigor. To improve the comparability of our sampling effort, we have stated the total person-hours invested in each of the three forest zones (144 person-hours per site) in the Collection method section. We appreciate the suggestions for future work and will consider incorporating plot-based methods in subsequent studies.

Reviewer Comment 2 - Statistical Analyses

“The correlation between altitude and species richness (R² = 0.86, p < 0.001) is robust, but with only three data points (n = 3), the test lacks inferential validity. Please rephrase this as a descriptive trend rather than a statistically significant correlation. Additionally, the authors could consider rarefaction or diversity indices (Shannon, Simpson) to better capture within-site heterogeneity."

Authors response 2

We agree with the reviewer’s assessment. Accordingly, we have revised the text in the Results and Discussion to rephrase the correlation as a "strong descriptive trend”. And calculated and incorporated the Shannon and Simpson diversity indices for each zone into the Results section to provide a more nuanced representation of within-site diversity, as suggested.

Reviewer Comment 3 - Floristic Composition and Elevational Gradient

"The discussion correctly highlights the decline in richness with elevation, but it would benefit from integration with known altitudinal patterns of Andean vegetation (e.g., treeline dynamics, páramo-puna transitions). Consider citing comparative floristic gradients from Ecuador, Bolivia, or northern Chile."

Authors response 3

We thank the reviewer for this excellent suggestion to place our findings within a broader Andean context. We have, accordingly, revised the discussion of the altitudinal gradient to better integrate our results with established ecological patterns.

The revised paragraph now contextualizes our observed decline in species richness by explicitly comparing it with known patterns from both wet páramo and dry puna gradients across the Andes. Furthermore, we now interpret the high dissimilarity of our lowest elevation zone (Z3) as evidence of its position at a critical ecotonal boundary, incorporating elements from lower-elevation vegetation zones, as suggested. This change can be found in the third paragraph of the Discussion section.

Reviewer Comment 4 - Conservation Implications

"The paper convincingly demonstrates that Zone 3 (lowest elevation) concentrates endemic and threatened species. However, the discussion should explicitly address why: is this due to milder climate, soil fertility, or lower disturbance intensity? Incorporating local management perspectives or restoration priorities would strengthen the applied value."

Authors response 4

We thank the reviewer for encouraging us to go deeper into the mechanisms behind our conservation findings. We have substantially revised the discussion to explicitly address the potential drivers for the concentration of endemic and threatened species in Zone 3.

While a quantitative analysis of these drivers is beyond the scope of this floristic study, we now propose and discuss likely mechanisms based on the available evidence and established ecological literature: a milder microclimate, greater habitat heterogeneity due to riparian influence, and differences in disturbance regimes between the sites.

Importantly, as suggested, we have now explicitly linked these proposed drivers to specific, actionable local management and restoration priorities, thus strengthening the applied value of our work. This can be found in the fifth paragraph of the Discussion section.

Reviewer Comment 5 - Figures and Tables

"Figures 6-9 are informative but dense. Simplify labels and ensure legends are self-explanatory. Table 1 could be condensed into a supplementary file to improve readability."

Authors response 5

We thank the reviewer for the feedback. We have revised Figures 6–9 to improve their clarity, simplifying labels and making the legends more descriptive.

Regarding Table 1, while we appreciate the suggestion, we believe its inclusion in the main text is essential to the paper's contribution. As this study is fundamentally a floristic inventory, we consider the complete species list to be primary data that should be directly accessible to the reader within the main document. Therefore, we have opted to retain it in the main text to ensure the integrity and completeness of our results.

Reviewer Comment 6 - References

“Verify DOIs and consistency in journal names (e.g., Rev. peru. biol.Revista Peruana de Biología).”

Authors response 6

We thank the reviewer for the comment. We have verified DOIs for accuracy. The journal names have been formatted using standard abbreviations (e.g., "Rev. Peru. Biol.") in accordance with the journal's submission guidelines.

Round 2

Reviewer 1 Report

Comments and Suggestions for Authors

See my original comments

reviewer